

# Conservation implications of primate trade in China over 18 years based on web news reports of confiscations

Qingyong Ni[1,2,*], Yu Wang[1,*], Ariana Weldon[2], Meng Xie[3],
Huailiang Xu[3], Yongfang Yao[3], Mingwang Zhang[1], Ying Li[1], Yan Li[1],
Bo Zeng[1] and K.A.I. Nekaris[2]

[1] College of Animal Sciences and Technology, Sichuan Agricultural University, Chengdu, Sichuan, China
[2] Nocturnal Primate Research Group, Oxford Brookes University, Oxford, UK
[3] College of Life Sciences, Sichuan Agricultural University, Yaan, Sichuan, China
* These authors contributed equally to this work.

Corresponding author
Qingyong Ni,
niqingyong@hotmail.com

## ABSTRACT

Primate species have been increasingly threatened by legal and illegal trade in China, mainly for biomedical research or as pets and traditional medicine, yet most reports on trade from China regard international trade. To assess a proxy for amount of national primate trades, we quantified the number of reports of native primate species featuring in unique web news reports from 2000 to 2017, including accuracy of their identification, location where they were confiscated or rescued, and their condition upon rescue. To measure temporal trends across these categories, the time span was divided into three sections: 2000–2005, 2006–2011 and 2012–2017. A total of 735 individuals of 14 species were reported in 372 news reports, mostly rhesus macaques ($n$ = 165, 22.5%, *Macaca mulatta*) and two species of slow lorises ($n$ = 487, 66.3%, *Nycticebus* spp.). During the same period, live individuals of rhesus macaques were recorded 206 times (70,949 individuals) in the Convention on International Trade in Endangered Species of Wild Fauna and Flora Trade Database, whereas slow lorises were only recorded four times (nine individuals), indicating that the species originated illegally from China or were illegally imported into China. Due to their rescued locations in residential areas ($n$ = 211, 56.7%), most primates appeared to be housed privately as pets. A higher proportion of 'market' rescues during 2006–2011 ($\chi^2$ = 8.485, d$f$ = 2, $p$ = 0.014), could be partly attributed to an intensive management on wildlife markets since the outbreak of severe acute respiratory syndrome (SARS) in 2003. More than half (68.3%, 502 individuals) of the primate individuals were unhealthy, injured or dead when rescued. Thus, identification and welfare training and capacity-building should be provided to husbandry and veterinary professionals, as well as education to the public through awareness initiatives. The increase in presence of some species, especially slow lorises, with a declining population in restricted areas, also suggests the urgent need for public awareness about the illegal nature of keeping these taxa as pets.

## INTRODUCTION

Hundreds of wild animal species are traded both legally and illegally to satisfy the market for exotic pets (*Bush, Baker & Macdonald, 2014*). Many of these species are native

to tropical areas where catching them in the wild is economically more viable than captive breeding (*Rosen & Smith, 2010*). Although international trade is becoming widely documented (*Da Silva et al., 2016*; *Reuter & Schaefer, 2016*; *Nijman et al., 2017*), many exotic species are captured for the national pet trade, remaining in their countries of origin. The incidence of this trade is much more poorly documented as official recording mechanisms, such as CITES (Convention on International Trade in Endangered Species of Wild Fauna and Flora) Trade Database (UNEP World Conservation. Monitoring Centre, Cambridge, UK), are lacking, and often enforcement is limited. Illegal trade undermines the efforts of developing nations to manage their natural resources (*Rosen & Smith, 2010*). Unsustainable harvest of wild animals for the pet trade has already led to population decline and collapse of many species (*Da Silva et al., 2016*; *Svensson et al., 2016*). In addition, individuals in the illegal market are often handled and transported under appalling conditions, creating an animal welfare concern (*Reuter & Schaefer, 2016*; *Fuller et al., 2017*).

Wildlife trade is a growing concern for primates, a group of long-lived and slow-reproducing species. They are traded for consumption; biomedical research; for zoos, wildlife collections and the entertainment industry (*Kavanagh, 1983*; *Nijman, 2005*); as pets; for the sale of body parts (bodies, skins, hair and skulls) used in traditional medicine; as talismans and trophies; and for magical purposes (*Alves, Souto & Barboza, 2010*; *Nijman et al., 2011*). The CITES Trade Database from 2005 to 2014 reported a global primate trade of some 450,000 live individuals plus an additional 11,000 body parts. More than 430,000 individuals (93%) in this trade are Asian species (*Estrada et al., 2017*), and thus Southeast Asia is considered as a primate trade hotspot (*Nijman, 2010*; *Rosen & Smith, 2010*).

China is the second-most primate diverse country in Asia and nine species are considered endemic (*Roos et al., 2014*). In recent years, people's demand for wild animal products has grown substantially with the development of a consumer economy, and thus, China has become one of the world's largest consumers of wildlife products (*Zhang, Hua & Sun, 2008*). Primate trade of 537,480 live individuals was reported in China from 1975 to 2017 based on the CITES Trade Database. Eleven native primate species, including four macaques (*Macaca* spp.), two colobus (*Trachypithecus* spp.), two slow lorises (*Nycticebus* spp.) and three gibbons (*Hylobates* spp., *Hoolock* spp. and *Nomascus* spp.), were reported as having been illegally trafficked in China (*Li et al., 2010*; *Hu et al., 2011*; *Gao, Ma & Wang, 2012*; *Yin, Yu & Peng, 2016*).

China became CITES contracting party in 1981, requiring all internationally traded CITES-listed species to be accompanied by valid permits or certificates. The Law of Wild Animals Protection of the People's Republic of China, 1 March 1989 forbids the hunting, killing, trade, import or export of wild animals classified as rare or endangered unless under special circumstances (*Li & Wang, 1999*). Primates, except for newly described species, are included in the Red List of China's Vertebrates Designated for Legal Protection (Table 1).

Mass media is one of the principal arenas within which issues come to the attention of decision makers, interest groups and the public (*Barua, 2010*). Media attention promotes conservation of primates, along with the Internet gaining importance in global wildlife trade and changing perceptions towards threatened species (*Nekaris & Campbell,*
**Table 1 Number of rescuing events and rescued individuals of native primates in China during three periods based on web news search.**

| Species | Chinese name | Key items for searching | Conservation status[b] | | | No. of rescuing events (individuals) | | |
|---|---|---|---|---|---|---|---|---|
| | | | IUCN | CITES | NPWAs | 2000–2005 | 2006–2011 | 2012–2017 |
| *Nycticebus bengalensis* | 蜂猴 | '懒猴' or '蜂猴' | VU | I | I | 19(30) | 68(139) | 91(160) |
| *N. pygmaeus* | 倭蜂猴 | '懒猴' or '蜂猴' | VU | I | I | 11(46) | 29(51) | 40(61) |
| *Macaca mulatta* | 猕猴 | '猕猴' | LC | II | II | 1(1) | 6(20) | 50(144) |
| *M. cyclopis* | 台湾猕猴 | '猕猴' | LC | II | I | 0 | 0 | 5(7) |
| *M. leucogenys* | 白颊猕猴 | '猕猴' | VU | – | – | 0 | 0 | 0 |
| *M. leonina* | 北豚尾猴 | '豚尾猴' or '平顶猴' | VU | II | I | 0 | 0 | 2(6) |
| *M. munzala* | 达旺猴 | '达旺猴' | EN | II | – | 0 | 0 | 0 |
| *M. assamensis* | 熊猴 | '熊猴' | NT | II | I | 0 | 1(1) | 7(7) |
| *M. thibetana* | 藏酋猴 | '藏酋猴' or '藏猕猴' | NT | II | II | 0 | 1(1) | 13(15) |
| *M. arctoides* | 短尾猴 | '短尾猴' or '红面猴' | VU | II | I | 0 | 2(2) | 10(11) |
| *Rhinopithecus roxellana* | 川金丝猴 | '金丝猴' or '仰鼻猴' | EN | II | I | 1(1) | 2(2) | 4(6) |
| *R. bieti* | 滇金丝猴 | '金丝猴' or '仰鼻猴' | EN | II | I | 1(1) | 0 | 1(1) |
| *R. brelichi* | 黔金丝猴 | '金丝猴' or '仰鼻猴' | EN | II | I | 0 | 0 | 0 |
| *R. strykeri* | 缅甸金丝猴 | '金丝猴' or '仰鼻猴' | CR | I | – | 0 | 0 | 0 |
| *Semnopithecus schistaceus* | 长尾叶猴 | '长尾叶猴' | LC | I | I | 0 | 0 | 1(2) |
| *Trachypithecus shortridgei* | 萧氏叶猴 | '叶猴' | EN | I | I | 0 | 0 | 0 |
| *T. pileatus* | 带帽叶猴 | '叶猴' | VU | I | I | 0 | 0 | 0 |
| *T. phayrei*[a] | 菲氏叶猴 | '叶猴' | EN | II | I | 0 | 0 | 0 |
| *T. crepusculus*[a] | 印支灰叶猴 | '叶猴' | EN | II | I | 0 | 0 | 0 |
| *T. poliocephalus* | 白头叶猴 | '叶猴' | CR | II | I | 0 | 0 | 0 |
| *T. francoisi* | 黑叶猴 | '叶猴' | EN | II | I | 1(1) | 2(2) | 1(14) |
| *Hoolock tianxing* | 高黎贡白眉长臂猿 | '长臂猿' | CR | – | – | 0 | 0 | 1(1) |
| *Hylobates lar* | 白掌长臂猿 | '长臂猿' | EW | I | I | 0 | 0 | 1(2) |
| *Nomascus leucogenys* | 北白颊长臂猿 | '长臂猿' | EW | I | I | 0 | 0 | 0 |
| *N. nasutus* | 东黑冠长臂猿 | '长臂猿' | CR | I | I | 0 | 0 | 0 |
| *N. concolor* | 西黑冠长臂猿 | '长臂猿' | CR | I | I | 0 | 0 | 0 |
| *N. hainanus* | 海南长臂猿 | '长臂猿' | CR | I | I | 0 | 0 | 0 |

**Notes:**
[a] The two species shared the same data since they were separated recently.
[b] Conservation status. IUCN red list category: CR, critically endangered; EN, endangered; VU, vulnerable; NT, near threatened; LC, least concern; EW, extinct in the wild; CITES Appendix I and II; NPWAs, National Protected Wild Animals Category I and II.

2012; Roberge, 2014). Public knowledge concerning wildlife conservation can be quantified by analyzing comments and associated data posted online. Here, we aimed to measure the number of species of traded primates in news reports found by or surrendered to authorities in China, and examine trends over time and differences among regions. Furthermore, we examined public statements in the reports to evaluate how well the public could identify species in comparison with official identification in these same reports; if members of the public knew whether or not species were threatened; and also evaluated health and welfare status of the rescued or confiscated animals. These data are critical to recognize the magnitude and diversity of illegally traded primates in China, and generate suggestions for management strategies and law enforcement.
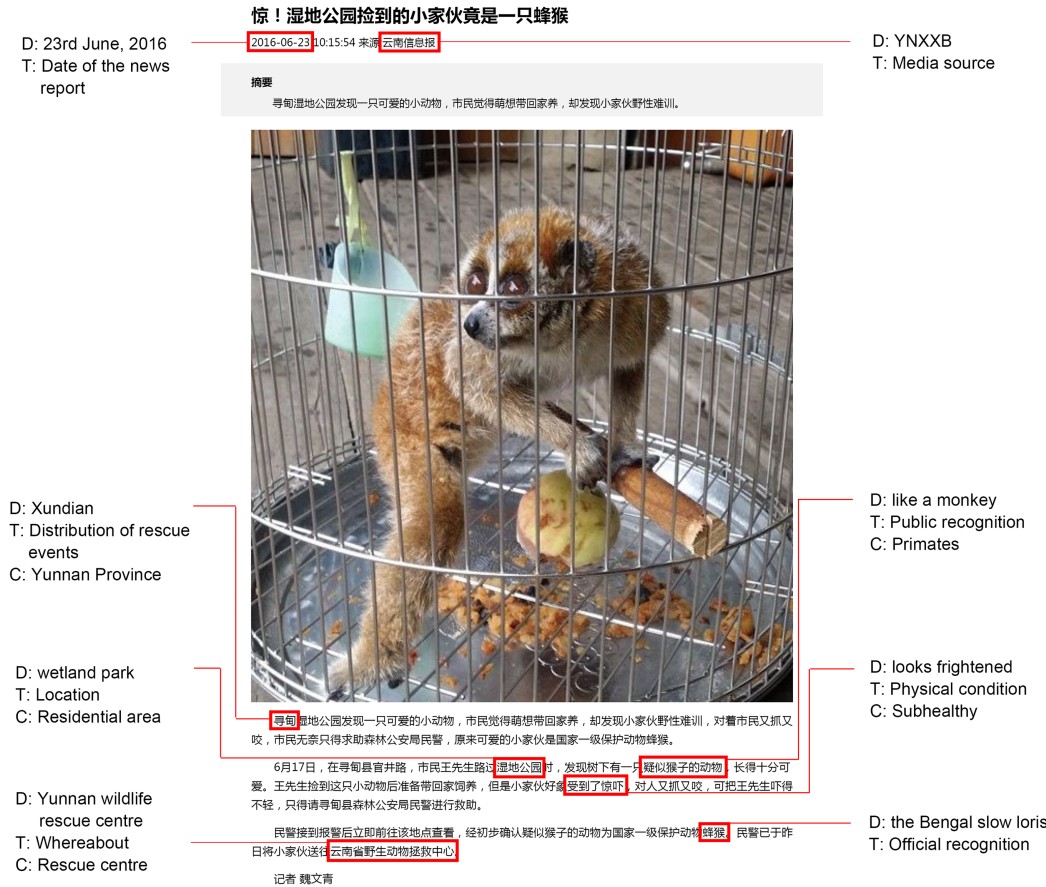

惊！湿地公园捡到的小家伙竟是一只蜂猴

**D: 23rd June, 2016**
**T: Date of the news report**

**D: YNXXB**
**T: Media source**

**D: Xundian**
**T: Distribution of rescue events**
**C: Yunnan Province**

**D: like a monkey**
**T: Public recognition**
**C: Primates**

**D: wetland park**
**T: Location**
**C: Residential area**

**D: looks frightened**
**T: Physical condition**
**C: Subhealthy**

**D: Yunnan wildlife rescue centre**
**T: Whereabout**
**C: Rescue centre**

**D: the Bengal slow loris**
**T: Official recognition**

**Figure 1 A typical example of data collection from a Chinese online news report of a rescue event.** D: Description in the news report; T: Type of the news description; C: Category of the description type. We identified the rescued animal in this news report as the pygmy slow loris based on the photograph, and thus the official recognition (the Bengal slow loris) was considered to be incorrect. Photo credit: Yunnanxinxibao (http://yn.xinhuanet.com/2016hot/20160623/3225860_m.html).

# METHODS

To reveal temporal variations in trade of native primate species in China, we used purposive sampling (*Newing et al., 2011*) to collect rescue or confiscation related news online. We considered rescuing or confiscating to be descriptions of primates surrendered to or confiscated by the authorities, hereafter referred to as rescue events. We conducted the searches in February 2018 and limited the period from 1st January 2000 to 31st December 2017, in three popular Chinese Web 2.0 search engines: Baidu, 360 and Bing. Baidu, especially, is by far the largest search engine in China, fulfilling a similar function to Google. Based on the Chinese name of each species, we entered manually the simplified Chinese key terms into each search engine (Table 1). We used '新闻'(news) category to select news articles and filtered the articles related to rescue events using keywords '救护' (rescue) or '查获' (seize) or '没收' (confiscate). Given the effects of search engine algorithms and previous search history on the results, we expected the potential bias could be reduced by cross validation of the three search engines. We combined all the news reports and excluded repetitive news based on date, site and media source.

**Table 2 Categories into which we placed contents, and example descriptions in the news reports for each type.**

| Type | Category | Descriptions in the reports |
|---|---|---|
| Physical conditions | Healthy | 'Lively', 'healthy', 'normal physical', 'have a good mental condition', 'No injuries and normal eating' |
| | Unhealthy | 'Hair loss', 'dermatopathya', 'unmoved, shivering, or full of fear', 'weak', 'undernourished' |
| | Injured | Visible wounds on the body |
| | Dead | Died during rescuing; corpse |
| Public recognition | Species | Mention its Chinese name |
| | Primate | 'Like a monkey' |
| | Unrecognized | 'Animal' 'can't recognize' |
| Whereabouts | Zoo | Be sent to a zoo or park |
| | Wildlife rescue centre | Be sent to a wildlife rescue centre |
| | Wild | Release to a nature reserve, forest area or suitable habitat area |
| | Unreported | No related statements |
| Location | Field | On the tree or ground near the forest |
| | Residential area | In the building or on the road of residential area |
| | Transporting | In the process of transportation, such as vehicles |
| | Market | Bird and flower market, agricultural markets, or pet shops |

Each report included various identification of the species included in the rescue event. These identifications were made by the public (public recognition), or an official who carried out the rescuing event, which was considered the official identification (Fig. 1). We categorized the public recognition as unrecognized; primate (but not to species); or species level identification (Table 2). Based on information provided in the news reports, especially photographs, we assessed the taxonomic status to compare with the official identification. Frequency in different categories of public recognition and accuracy of official identification were used as proxies for public knowledge. We followed the primate taxonomy as listed in *The Handbook of the Mammals of the World*, Volume 3 (Primates), and original accounts for two taxa not included in that resource (*M. leucogenys—Li, Zhao & Fan, 2015*; *Hoolock tianxing—Fan et al., 2017*).

We collected information in each news report on date of rescue or confiscation, number of individuals, location of rescue (e.g. field, market, residential area or transporting vehicles), physical condition of rescued individuals (e.g. healthy, unhealthy, injured or dead), and whereabouts of the individuals after being rescued (e.g. zoo, wildlife rescue centre, released into wild, or unreported) (Table 2; Fig. 1). China's primates are naturally distributed in 21 of 34 provincial-level administrative units (PLAUs), with four provinces in west and southwest China (Yunnan = 15 species, Guangxi = eight species, Tibet = eight species, Guizhou = six species), containing the highest diversity (Fig. 2A). We also recorded provinces where news was/had been reported to determine distribution of rescuing (Fig. 1).

For an overview of international trade, we examined data from the CITES Trade Database, which provided all records of import, export and re-export of CITES-listed species. The data were downloaded in May 2018 and 'year range' were limited from 2000 to 2016 with 2016 being the last year for which data were available. We searched live (LIV) animals in Order Primates traded with all sources and purposes, and focused on 27

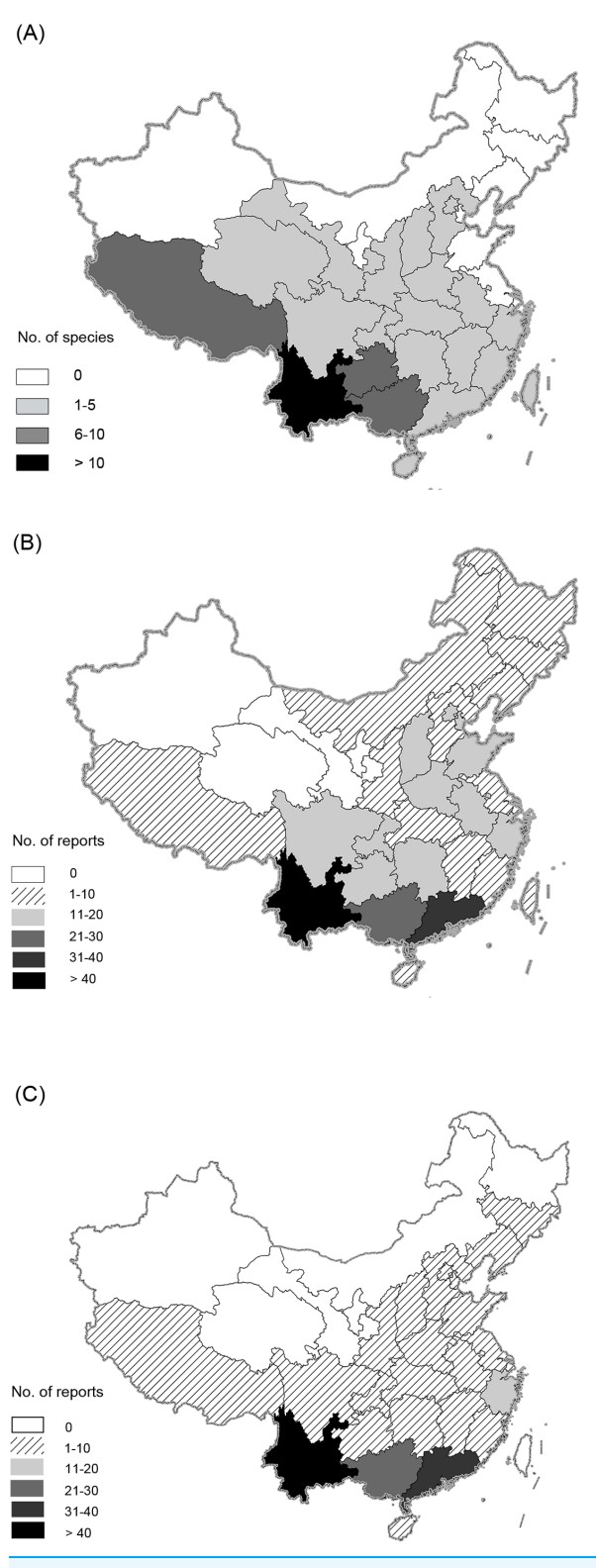

**Figure 2 Distribution of native primate species (A), and rescuing reports of primates (B) and slow lorises (C) across provinces in China.**

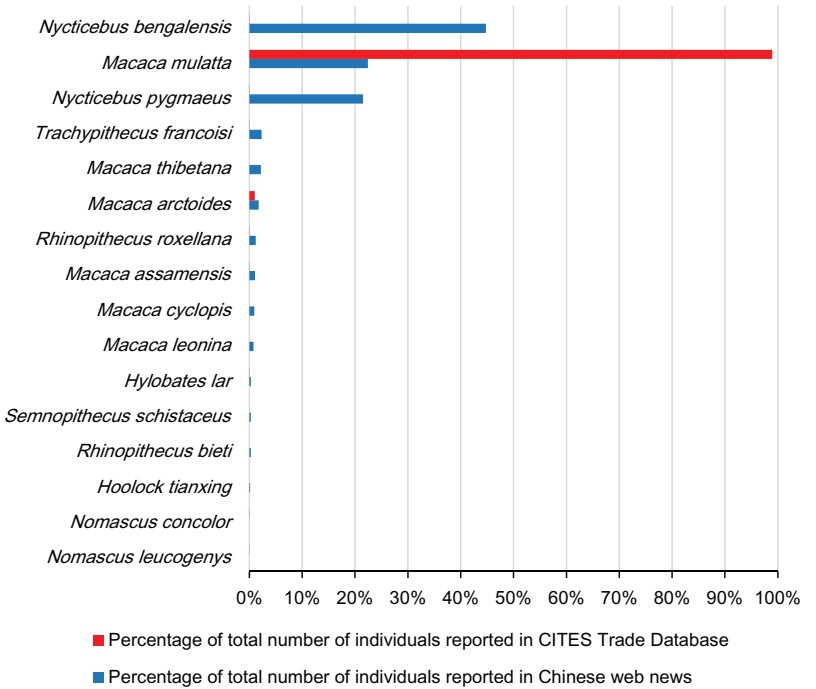

*Nycticebus bengalensis*
*Macaca mulatta*
*Nycticebus pygmaeus*
*Trachypithecus francoisi*
*Macaca thibetana*
*Macaca arctoides*
*Rhinopithecus roxellana*
*Macaca assamensis*
*Macaca cyclopis*
*Macaca leonina*
*Hylobates lar*
*Semnopithecus schistaceus*
*Rhinopithecus bieti*
*Hoolock tianxing*
*Nomascus concolor*
*Nomascus leucogenys*

0%  10%  20%  30%  40%  50%  60%  70%  80%  90%  100%

■ Percentage of total number of individuals reported in CITES Trade Database
■ Percentage of total number of individuals reported in Chinese web news

**Figure 3 Percentage of the total number of individuals per each indigenous primate species reported in CITES Trade Database during 2000–2016 and Chinese web news during 2000–2017 in China.**

indigenous primate species in China (Table 1). The data implicating China as importer and exporter were combined to obtain number of individuals traded per year for each species.

We divided the time span from 2000 to 2017 into three sections: 2000–2005, 2006–2011 and 2012–2017, and used the Kruskal–Wallis non-parametric test to examine variations over periods in rescuing frequency. To measure temporal trends of public knowledge about primate conservation, Kruskal–Wallis test was also used to compare the proportion of each description category in three time sections towards those species which were reported in more than 6 years. We calculated the Shannon–Wiener Index ($H = -\sum_{i}^{S} p_i \ln p_i$) and Pielou Index ($E = H / \ln S$) for each year to evaluate diversity and evenness of primates reported, where $S$ = total number of species recorded in a given year, $p_i$ = the proportion of individuals belonging to $i$th species. Spearman's Rank Correlation Coefficient was used to analyse annual variations of diversity and evenness. All the tests were two-tailed and a threshold for significance was $p < 0.05$.

## RESULTS

### Temporal variations of primate rescuing frequency in China

We filtered 372 valid news reports based on the topics of rescuing and confiscation, including 735 individuals of 14 primate species (Table 1; Fig. 3). The Bengal slow loris (*Nycticebus bengalensis*) was the most reported species with 329 (44.8%) individuals, followed by the rhesus macaque (*M. mulatta*, 165 individuals, 22.5%) and the pygmy slow

loris (*N. pygmaeus*, 158, 21.5%), while 13 of 27 primate species distributed in China were never reported (Fig. 3). We recorded rescue events of Bengal and pygmy slow lorises every year during the 2000–2017 period. Rhesus macaque rescue events were reported in 10 years (2005, 2007, 2010–2017) and the Tibetan macaque (*M. thibetana*) in recent 7 years (2011–2017). The rescue news related to other species, however, was individually reported in no more than 6 years. The diversity index increased significantly over time (Spearman's rank Correlation Coefficient, $\rho = 0.862$, $p < 0.001$, $N = 18$), as well as evenness ($\rho = 0.488$, $p = 0.040$, $N = 18$). Primate rescue frequency tended to increase during the last 6 years from 2012 to 2017 (Fig. 4A) while number of news reports specifically on slow lorises fluctuated between years (Fig. 4B).

Comparatively, an average of 4,219 ± 1,618 live individuals of native primate species per year, including ten species in total, were recorded in CITES Trade Database from 2000 to 2016 (Fig. 4C). The rhesus macaque contributed most of these internationally traded individuals (70,949, 98.6%, 206 records), followed by the stump-tailed macaque (*M. arctoides*, 726, 1.0%, three records). Nine individuals (four records) of slow lorises and none of Tibetan macaques and were reported in CITES Trade Database over the studied period.

## Location and provincial distribution of rescuing news reports

Among 372 rescue events recorded, 211 (56.7%) reports were located in residential areas, followed by 70 (18.8%) in wild areas, 54 (14.5%) during transporting and 37 (9.9%) in the market. The proportion of market rescues was significantly higher in 2006–2011 than the other time sections but fewer individuals were rescued from markets in 2000–2005 based on the reports ($\chi^2 = 8.485$, $df = 2$, $p = 0.014$), especially for Bengal slow lorises ($\chi^2 = 11.832$, $df = 2$, $p = 0.003$) and rhesus macaques ($\chi^2 = 9.544$, $df = 2$, $p = 0.008$).

Primate rescuing news covered more than 190 counties in 29 PLAUs throughout China (Fig. 2B), with a considerable proportion of rescuing events (130, 36.9%) taking place in Yunnan province, followed by Guangdong (33, 9.4%) and Guangxi (26, 7.4%). The rescuing news related to slow lorises occurred in 26 PLAUs (Fig. 2C), while rhesus macaques rescues was reported in 21 PLAUs, and other species were not individually reported in more than seven PLAUs. It is noteworthy that data from Taiwan (4), Hong Kong (1) and Macau (0) were limited due to unpopular use of simplified Chinese.

## Physical conditions and whereabouts of individuals rescued

We found that more than half (68.3%, 502 individuals) of the primate individuals were unhealthy, injured or dead when rescued. Of 105 individuals whose injuries were specified, most (40, 38.1%) were suffering from leg wounds. The proportion of healthy individuals rescued was significantly lower in 2006–2011 than the other two time periods ($\chi^2 = 6.140$, $df = 2$, $p = 0.046$).

The percentage of healthy individuals varied significantly over time in Bengal slow lorises ($\chi^2 = 6.579$, $df = 2$, $p = 0.037$) and Tibetan macaques ($\chi^2 = 9.563$, $df = 2$, $p = 0.008$), as well as the percentage of injured individuals in pygmy slow lorises ($\chi^2 = 8.503$, $df = 2$, $p = 0.014$), rhesus macaques ($\chi^2 = 8.812$, $df = 2$, $p = 0.012$) and Tibetan macaques

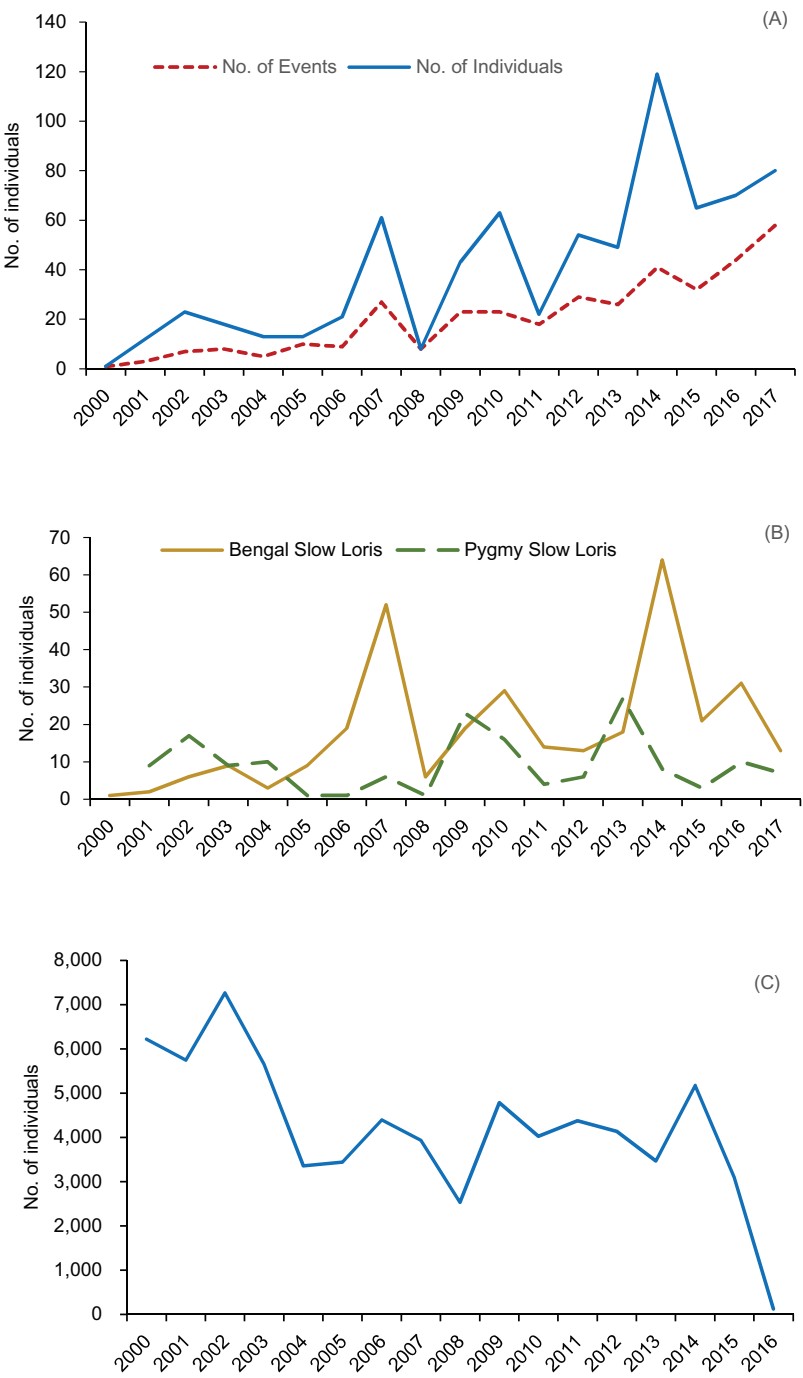

**Figure 4 Annual trends in traded or rescued primate individuals reported by CITES and web news.** (A) Annual number of events and individuals reported in Chines web news. (B) Annual number of individuals of slow lorises reported in Chinese web news. (C) Annual number of individuals of indigenous primates reported in CITES trade database.

($\chi^2$ = 9.563, d$f$ = 2, $p$ = 0.008). The whereabouts of the individuals after being rescued were often unreported (116, 31.2%), followed by 'wildlife rescue centre' (91, 24.5%), field (86, 23.1%) and zoo (79, 21.2%).

**Table 3 Frequency in different categories of public recognition ($n$ = 372) and accuracy of official recognition based on taxonomic assessments of photographs provided by the news reports ($n$ = 220).**

| Species | Public recognition | | | Official recognition | |
|---|---|---|---|---|---|
| | Species | Primates | Unrecognized | Correct | Incorrect |
| Nycticebus bengalensis | 68 | 42 | 68 | 85 | 0 |
| Nycticebus pygmaeus | 30 | 20 | 30 | 26 | 26 |
| Macaca leonina | | 2 | | 1 | 1 |
| Macaca mulatta | 9 | 46 | 2 | 41 | 3 |
| Macaca assamensis | 1 | 7 | | 6 | 0 |
| Macaca cyclopis | | 5 | | 4 | 0 |
| Macaca thibetana | | 12 | 2 | 6 | 2 |
| Macaca arctoides | | 12 | | 9 | 0 |
| Rhinopithecus roxellana | 1 | 6 | | 4 | 0 |
| Rhinopithecus bieti | 1 | 1 | | 1 | 0 |
| Semnopithecus schistaceus | | 1 | | 0 | 0 |
| Trachypithecus francoisi | | 4 | | 3 | 0 |
| Hylobates lar | 1 | | | 1 | 0 |
| Hoolock tianxing | 1 | | | 1 | 0 |

## Species recognition

The individuals in nearly half of rescuing events (158, 42.5%) were recognized as primates by the public, and the individuals in 112 events (30.1%) were recognized as a specific species (Table 3). The public could not recognized the animals or referred to primates in 102 events (27.4%). The public recognitions of the Bengal slow loris was consistent with the pygmy slow loris over the three time periods. The proportion of rescuing events in which the individuals could be recognized to a species level by the public varied over time for the rhesus macaque ($\chi^2$ = 6.733, d$f$ = 2, $p$ = 0.035) and the recognition percentage of individuals identified as 'primates' varied significantly for the Tibetan macaque ($\chi^2$ = 9.389, d$f$ = 2, $p$ = 0.009).

The official recognition was reported to species level in all of the news reports, but species in 14.5% (32/220) of news including photos were incorrectly identified (Table 3). The pygmy slow loris, which was usually recognized as the Bengal slow loris, contributed most (26/32, 81.3%) to these wrong identifications. All the other incorrect identifications (6) were related to species of Genus *Macaca* (Table 3), especially for rhesus macaques (3) and Tibetan macaques (2).

## DISCUSSION

### Public knowledge towards illegal primate trade in China

For certain native primate species in China, few individuals were traded internationally based on the CITES Trade Database, whilst rescuing or confiscating news reports revealed that they were frequently traded in domestic areas. In addition, the lower frequency of rescuing or confiscating and a focus on web news mean that the number of individuals

traded might be underreported. Thus, it could be argued that a large amount of illegal trade at national level, especially for Bengal and pygmy slow lorises, appeared to be underrepresented by official data. Rhesus macaques composed a large proportion of rescuing reports, which was consistent with the fact that it is the most abundant primate species in China, and widely traded or housed for biomedical purpose (*Bontrop, 2001*; *Fan & Song, 2003*). Given the extensive captive breeding throughout China (*Fan & Song, 2003*), a large number of animals of rhesus and Tibetan macaques rescued may be originally from captive populations. Without any breeding centre in China, not to mention internationally, probably all slow lorises were wild-captured and trafficked illicitly.

More than half of the primates in the rescuing news were located in residential areas, indicating that they had probably escaped from households where they were kept as pets. Furthermore, in 65% of reports only a single individual was rescued or confiscated, underlining that the animals were the end-point of trade chain, and had presumably been trafficked several times before being housed (*Duarte-Quiroga & Estrada, 2003*). The Chinese government has markedly strengthened management of wildlife markets since the outbreak of SARS in 2003, which was considered to originate from small wild animals (*Zhong, 2004*), likely explaining the significant increase of primates rescued from markets during 2006–2011. Illegal trade related to wildlife markets has declined during recent years, and large specialized traditional open markets tend to be replaced by underground trade networks, in particular, the booming online trade on social media (*Xiao, Guan & Xu, 2017*). Eight online transactions of slow lorises, were detected and penalized based on web news from 2011 to 2017, but only two reports before 2010s.

## Spatial variation in rescuing frequency associated with wildlife trade

Frequency of rescuing news on primate species varied remarkably between PLAUs, which indicates a significantly heterogeneous illegal trade distribution across China. A bulk of rescuing events took place in southwestern PLAUs, including Yunnan, Guangxi, Guizhou, Tibet and Sichuan. This is consistent with the highest primate diversity in this area, where more than 92% of the total species in China are distributed and 78% are endemic. In addition, these areas are situated near Southeast Asia, which is a hotspot for global biodiversity (*Myers et al., 2000*; *Sodhi et al., 2010*) and wildlife trade (*Nijman, 2010*). Yunnan and Guangxi, in particular, share long borders with Vietnam, Myanmar and Laos, and are considered as one of the major entrances for wildlife trafficking from neighbouring nations (*Li & Li, 1998*; *Shepherd & Nijman, 2007*; *Zhang, Hua & Sun, 2008*).

Guangdong province is one of the main destinations for smuggling and the largest wildlife markets (*Zhang, Hua & Sun, 2008*; *Chow, Cheung & Yip, 2014*), making it another possible hotspot of primate trade. Along with Guangdong, Beijing, Shandong and Zhejiang are among the most developed PLAUs in China and contributed a lot to the illegal wildlife trade (*Li & Lu, 2014*; *Yu et al., 2017*). *Zhang & Yin (2014)* found that consumers with higher income background had a higher wildlife consumption rate, suggesting that financial strength increases people's propensity to consume wild

animals. To support this point, few primate rescues were reported in north-western PLAUs, the less developed regions in China. The lower primate trade rate observed in northwest and northeast may also result from a long distance from source areas. With the expansion of online trade in recent years, the trafficking sites have become increasingly extensive and scattered, and the distance between sources and the point of retail tend to be greater (*Zhang, Hua & Sun, 2008*).

## Challenges of welfare and captive management in primate rescuing

Primate individuals were mostly sent to zoos, rescue centres or released into the wild after being rescued. For a considerable number of animals (116, 31.2%), we were unable to extrapolate their final destinations from the news reports. Given the scattered sites, it was not surprising that all the individuals were rescued by local forestry staff, who might encounter difficulties during rescuing, such as species identification. Lack of discrimination in the trade, especially in morphology, combined with unresolved taxonomic issues, impedes assessing each taxon's potential vulnerability to trade (*Vonk & Wüster, 2006*; *Nekaris & Jaffe, 2007*; *Nekaris & Nijman, 2007*). Genetic, vocalization and behavioural analyses are essential for rescue and release programmes, yet may be beyond the capabilities for some facilities (*Mootnick, 2006*). As a consequence, the pygmy slow loris was usually confused with the Bengal slow loris, and the two species were thus housed and released indiscriminately. Limited understanding of slow lorises taxonomy throughout their distribution ranges confounded attempts to reintroduce these animals, or hold and breed them in captive facilities (*Nekaris & Starr, 2015*).

Primates have specific physiological, physical, social and nutritional requirements, and it is unlikely that the welfare of pet animals can be adequately addressed in normal households (*Soulsbury et al., 2009*). Captive primates, including those in zoos and rescue centres, have been commonly observed to suffer from incorrect diet, wounds or disease, unnatural environment, and fear or distress (*Duarte-Quiroga & Estrada, 2003*; *Hevesi, 2005*; *Nekaris et al., 2010*). The specialist needs of primate species also mean that they might experience elevated mortality and perish quickly in captivity (*Fitch-Snyder, Schulze & Larsson, 2000*). More than one third of individuals of slow lorises, died within the first 6 months in rescue centres in southern Yunnan (Q. Ni, 2017, personal communication). Hence, for many rescue institutions, immediate re-release is often considered as preferable (*Nekaris & Jaffe, 2007*). The problem of rehabilitating captive animals without regard to genetic and ecological assessments, geographic distribution, and monitoring has become another difficult issue that remains to be resolved (*Duarte-Quiroga & Estrada, 2003*; *Nekaris & Starr, 2015*). The individuals in all the 41 news reports related to release were reintroduced into wild directly during 18-year periods, by the local authorities without preparation and training involved, indicating that hard release has been rampant in primate rescuing throughout China. This might lead to high mortality of animals released (*Moore & Nekaris, 2014*) or endanger wild populations and other animals with disease transmission (*Wallis & Lee, 1999*), thus becoming a useless conservation plan.

## Implications for primate conservation in China

Accurate measures of wildlife trade are essential to devising sound conservation decisions, yet collection and quality control of such data are challenging (*Thomas & Albert, 2006*). The database generated by CITES offers an unparalleled opportunity to analyse international trade in species of conservation concern (*Foster, Wiswedel & Vincent, 2016*). However, the present official statistics have limited capabilities in representing species illegally harvested and traded (*Phelps & Webb, 2015*), especially for those trafficked at national or regional level. Therefore, taking these data at face value can sometimes distort the perceived risk of wildlife exploitation and lead to misallocation of resources and ineffective conservation efforts (*Thomas & Albert, 2006*; *Robinson & Sinovas, 2018*). As a case study, the discrepancy between few CITES trade records and massive rescuing news reports of slow lorises in China emphasize a need of more reliable and comprehensive understanding of some species, and calls for a harmonized mechanism in estimating national and regional wildlife trade.

The success of wildlife conservation largely depends on public perspective, as well as assessment of causes that influence their outlook (*Wilson & Tisdell, 2007*; *Lindemann-Matthies & Bose, 2008*). Media reports indicate that public knowledge towards primates in China varied between species. Macaques, along with leaf monkeys and snub-nosed monkeys, represented the typical image of 'monkeys' (i.e. primates), in a broad sense, and were well known for the general public in China. The famous Monkey King, Sun Wukong, for instance, was considered to be originated from these species (*Qin, 2010*). Similarly, though mysterious in deep forests, gibbons historically occurred in many poems and paintings and were rich in symbolic meanings (*Geissmann, 2008*; *Zhang, 2015*), whereas slow lorises played a minimal role in Chinese culture. Few relevant publications, combined with limited distribution areas in the southwest, has resulted in them being the least known primates within the country. Chinese people usually judge animals by ethical standards and emphasize the creature's usefulness to humans, but ignore the physical characteristics of the animals (*Zhang, 2015*), leading to a series of misunderstanding on slow lorises. One of slow lorises' perceived uses to treat epilepsy—called 'mad sheep disease' by local communities—in traditional Chinese medicine, as a result, was mostly attributed to confusing Chinese common names with similar pronunciation (feng) among '疯' (mad)-, '风' (wind)- and '蜂' (bee)-猴 (monkey).

Given the clandestine illegal trade of primates based on web news reports, it can be concluded that monitoring systems of wildlife trade within China are insufficient, and there is an urgent need for initiatives to make regulatory mechanisms more effective (*Zhang, Hua & Sun, 2008*; *Nijman, 2010*). A common problem in the enforcement of legislation to protect animals from illegal trade is the inability to identify species due to inadequate funding, education and staffing. Recommendations to address these areas should include identification-training initiatives and capacity-building work (*Li & Wang, 1999*). In addition, it is highly recommended that an approach concerning awareness initiatives and education programmes should be developed towards the public to make them more conscious about the illegal wildlife trade, with the final intention of discouraging the consumers to buy wildlife products.

By 2008, 40 primate-breeding centres in China contained over 40,000 individuals, mostly rhesus macaques, and the number has been steadily increasing in recent years (*Jiang et al., 2015*; *Cyranoski, 2016*). It is necessary to strengthen captive management and improve animal welfare, which was still inconsistent and rudimentary, since the concept has been introduced into mainland China only in the last few decades (*Lu, Bayne & Wang, 2013*). The authorities should accelerate the legislative process and provide animal welfare education to the public, as well as training to husbandry and veterinary professionals.

The demand for pet primates, together with habitat loss and fragmentation, exerts a significant pressure on wild populations. In particular, slow lorises are perceived as suitable pets by both buyers and sellers due to their 'cute and cuddly' appeal, and have been one of the most popular primate taxa in wildlife markets (*Nekaris, 2014*). The widespread illegal trade in China seems to be incompatible to their restricted distributions, high threat category, and poor public knowledge. Taking into account their small and declining wild population, it is urgent to take actions for conservation of this neglected and threatened primate taxon.

## CONCLUSION

We have presented a novel data on primate trade within China based on web news reports regarding rescuing or confiscating. The results indicate that some native primate species, particularly Bengal and pygmy slow lorises, are threatened by domestic illegal trade, which appears to be 'unrecognized' in official channels, and lack of public knowledge impedes efforts to conserve these species effectively. In spite of potential bias in search results caused by search engine algorithms and manual filtering, and lack of the firsthand data from authorities, zoos or wildlife rescue centres, we expect that this study could facilitate the initial steps to raise public awareness on primate trade in China, especially for slow lorises.

## ACKNOWLEDGEMENTS

We are grateful to Vincent Nijman, Marika Roma, Kathleen Reinhardt, Cristy Jasso, Miguel Guinea and Eleonora Favilli for their comments on the manuscript.

### Funding
This study was supported by the National Natural Science Foundation of China (No. 31501873) and Kadoorie Farm & Botanic Garden. The funders had no role in study design, data collection and analysis, decision to publish, or preparation of the manuscript.

### Grant Disclosures
The following grant information was disclosed by the authors:
National Natural Science Foundation of China: 31501873.
Kadoorie Farm & Botanic Garden.

## Competing Interests

The authors declare that they have no competing interests.

## Author Contributions

- Qingyong Ni conceived and designed the experiments, performed the experiments, analysed the data, prepared figures and/or tables, authored or reviewed drafts of the paper, approved the final draft.
- Yu Wang performed the experiments, analysed the data, contributed reagents/materials/analysis tools, prepared figures and/or tables, approved the final draft.
- Ariana Weldon authored or reviewed drafts of the paper, approved the final draft.
- Meng Xie performed the experiments, analysed the data, approved the final draft.
- Huailiang Xu contributed reagents/materials/analysis tools, approved the final draft.
- Yongfang Yao contributed reagents/materials/analysis tools, approved the final draft.
- Mingwang Zhang contributed reagents/materials/analysis tools, approved the final draft.
- Ying Li contributed reagents/materials/analysis tools, approved the final draft.
- Yan Li contributed reagents/materials/analysis tools, approved the final draft.
- Bo Zeng contributed reagents/materials/analysis tools, approved the final draft.
- K.A.I. Nekaris authored or reviewed drafts of the paper, approved the final draft.

## Data Availability

The raw data is provided in the Supplementary Files 1 and 2. The data shows preliminary statistics of news reports regarding primate rescuing or confiscating in China, and CITES reports on primate trade implicating China.

## Supplemental Information

Supplemental information for this article can be found online at http://dx.doi.org/10.7717/peerj.6069#supplemental-information.

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
