# Peer review of "Conservation implications of primate trade in China over 18 years based on web news reports of confiscations"

_PeerJ, doi:10.7717/peerj.6069_

## Round 0.1 · original submission · Major Revisions

Many thanks for submitting this very interesting study. The reviewers and I are all enthusiastic about this work however currently the method section is insufficient making it difficult to determine if the results and therefore conclusions are valid. I would therefore encourage you to look at the reviewers comments, which I think are helpful, and in particular significantly expand on the methods you use so that they are reproducible.

Reviewer 1 ·

Basic reporting

Study is generally well designed and the manuscript is nicely developed and structured. Overall, clear and unambiguous, professional English used throughout, but here are many problems with language throughout the manuscript, and I suggested below that it needs editing by a native English speaker. Manuscript is adequately supported by references, and sufficient field background/context is provided. Manuscript also has professional article structure, with adequate figs and tables, and raw data was provided for review (I am not sure if it is going to be published online in some data repository though). Manuscript is also self-contained with relevant results to hypotheses.

Experimental design

The study represents original primary research that fits Aims and Scope of the journal. Research question well defined, relevant and meaningful, and the authors stated how research fills an identified knowledge gap. Investigation seems to be performed to a high technical & ethical standard. However, methods are not described with sufficient detail and information, which does not allow for study replication or to judge the validity of the study. I have therefore suggested a major revision of the manuscript to address those issues.

Validity of the findings

Data in general appears to be robust, statistically sound and controlled. There is however insufficient information on the approach, so it was not possible to judge its validity fully. While general conclusions are adequately stated, there is certain level of over-interpretation of results, and conclusions drawn that were not supported by the study. I addressed this issue in my general comments below, and suggested a major revision of the manuscript.

Additional comments

In their manuscript titled "Public knowledge on primate trade in China over 18 years based on web news reports: Implications for urgent conservation actions of slow lorises", MS #30188, authors made an analysis of the online news articles dealing with national trade in primates in China, in order to assess the extent and impact of illegal trade, and related public awareness and knowledge. Study is novel and potentially highly relevant given the ongoing level of illegal wildlife trade worldwide, and especially given the lack of reliable information on primate trade on in China. Study is generally well designed and the manuscript is nicely developed and structured, and should be of interest for a wider readership.

There are however several issues that would have to be resolved before the manuscript can be considered as suitable for publishing. My main concerns are related to the Methods section, as the data collection and assessment approach were not described in sufficient detail, so it is not possible to evaluate validity of the study. I listed my main comments below. Manuscript should be also edited for language by a native English speaker.

Major comments:

1. The main focus of the paper seems unresolved, as the paper strays in different sections from addressing all primates to being focused on slow lorises. In Methods, web search terms represent all primates in China, not just slow lorises, and such focus is also confirmed in many sentences (e.g. Lines 121-122, "We focused on 27 indigenous primate species in China"), as well as in the statement about the study objective in Lines 84-91. As a reader I was confused by such shifting of focus from all primates to lorises and back, and the mention of slow lorises in the title, abstract and elsewhere in the text was misleading, as it does not seem supported by the rest of the manuscript. It might be better to make it clear in the paper, title and abstract that the focus is on all primates, but that the emphasis will be placed on lorises due to their prominence in illegal trade and their endangered status.
This should be addressed also in the title - either to keep a broad focus and state just "Implications for urgent conservation", or to revise manuscript and clarify and justify the focus on slow lorises. Furthermore, I think that in the title it would be more appropriate to use "for slow lorises" or "on slow lorises" than "of slow lorises".

2. There seems to be also a mix-up in the study regarding the metric that was used, number of rescue reports. Authors mainly claim that rescue reports represent an indication (a proxy) of the level or intensity of primate/loris trade. However, elsewhere they seem to use the same metric as a proxy of public knowledge and awareness of primate rescues, or as it was even more broadly described, "to measure public awareness about primate conservation" (Line 123). What does the amount and content of online news on primate rescues really indicate, trends in primate rescuing or trends in the public awareness on rescuing activities? It appears as the authors couldn't decide about this either - for example, in Lines 125-126 they said that the statistical tests were applied to "examine variations over periods in rescuing frequency and the proportion of public knowledge towards those species".
In my opinion it is a combination of both, frequency/amount of such news articles indicates both the presence of that activity (primate rescues) and public awareness and attention towards such activity - and, arguably they also represent interest of news agencies in such activities. This should be clarified in the text, perhaps also addressed as a methodological problem in Discussion, and use of the same metric for different interpretations should be avoided.

3. I would avoid equating number of news articles and public awareness on that topic (Line 123 and elsewhere), since the former does not have necessarily to result in the latter - presence of news articles (especially on Internet, since only a small portion of population has Internet access) does not have to result in or be proportional with public awareness.

4. It is left unclarified in the Methods whether the search was made by using some automatic method (such as the Google API is often used for Google search), or the terms were simply entered manually on the webpage of each search engine. If the latter was the case, authors should address in the Methods section the problem of personalized results provided by search engine algorithms and previous personal search history, as a potential source of bias in their results. Automatic search methods, such as the above mentioned Google API for Google search, have the advantage of providing depersonalized results (with no effects of personal search history, geographic location of the person doing the search, etc.) which prevents risk of this source of a bias and provides replicability of the study.

5. It is not clear from the Methods whether only simplified Chinese key terms, representing species names, were used to conduct web search, or some additional terms were used that represented rescue of confiscation activities. In order to allow replicability of the study, the exact search phrases entered in each search engine should be presented, either in the text or in a supplementary material.

6. Authors stated that they have selected news articles that were related to rescuing events. How was that done? This is one of critical steps in the methods, but it is not explained properly. Did the authors do this manually, by accessing and checking each news article separately? I don't expect that this was feasible, since it is mentioned in the text that there were millions of results obtained. Was there a second search step, used for filtering of results, prior to their manual analysis? If there was another search conducted within news articles obtained based on species names, exact search phrases and other relevant information should be provided. Authors mention filtering of results in Results section (Line 135, "we could filter 372 valid news reports based on the topics"), but leave it unexplained.

7. If authors had to deal with millions of news articles obtained through search engines, what was the stopping rule that was applied (i.e. criteria to decide at which point all relevant articles were identified, and further results are unlikely to be relevant)?

8. Authors used three search engines to obtain news articles. How were results from the three engines pooled together? This is left unexplained in the text.

9. I also have an issue with the "unknown" category used for different characteristics of news reports, and the way this is interpreted in the text, which sometimes seem to indicate that the author of the news article, general public or officers doing confiscation/rescuing were unable to determine the species, or its destination, etc. The way I understood it was that this category represents only situations where such information was missing from the news article, but that does not mean that the author of the article or persons involved in the described rescue event were not in possession of such information.
For example, in Lines 179-181 it is stated that "individuals in 102 events (27.4%) could not be recognized by the reporters or the public when rescued". How was this determined, whether reporters or even the general public was unable to recognize the species? The only thing that can be confirmed by the news article was that the species name was not disclosed by the journalist. Or the authors used some additional information to determine this? This goes on further in Lines 181-186, and the authors seem to mix-up the fact that the species name was not reported in a news article with the one that the species was actually not recognized. This issue will have to be clarified.

10. Results seem also to be over-interpreted - in the Discussion, authors state for primates in China that "rescuing or confiscating news reports revealed that they were notably threatened by domestic trade" (196-197). How was this estimated, based on which information or baseline value is the observed number of online news on rescue events used to make judgment about illegal trade as representing a notable extinction threat? I would also expect that the level of illegal wildlife trade is certainly an important threat, but this cannot be claimed based on the results of the study alone. Similarly, they claim that results indicate the number of reported individuals might be "just the tip of iceberg" (Lines 198-199), or that it is a "rampant domestic trade" (Line 287) or "rampant illegal trade" (Line 307) - although this is likely, it cannot be concluded based on presented results.
There are other statements in the Discussion that do are not supported by the results, so they should not be presented in that way, for example in Lines 220-224: "Some stable supply-and-purchase partnerships, including wildlife brokers and their patrons, should be responsible for most of these underground transactions. We observed also the presence of three reports regarding Internet fraud about primate sales, suggesting that owning primate pets was still attractive and worth taking a risk for some citizens." I would suggest either rephrasing or deleting these sentences.

11. There are many problems with language throughout the manuscript. I listed many such instances below, but the list is far from complete. Manuscript should be carefully checked and edited by a native English speaker.

12. Figure 4 is confusing, I could not understand what is exactly shown there, or what is represented by "market" and "healthy"


Other comments:

13. Lines 35-36 - I suggest replacing "education concerning awareness initiatives to the public" with "education to the public through awareness initiatives"

14. Line 40 - delete "in trade", since "trade is already mentioned in this sentence

15. Line 44 - replace "For many these species, they are" with "Many of these species are"

16. Line 45 - insert "national" before "pet trade"

17. Line 49 - replace "collapse for many species" with "collapse of many species"

18. Lines 53-54, "Primate trade is a growing concern for this group of long-lived slow-reproducing species" - I suggest rephrasing the sentence as "Wildlife trade is a growing concern for primates, a group of long-lived and slow-reproducing species"

19. Line 76 - replace "four" with "for"

20. Line 84 - delete "and" after the word "of"

21. Lines 85-86 - replace "examine trends which these data changed over time and varied between regions" with "examine trends over time and differences among regions"

22. Lines 89-91 - consider revising this sentence, to improve the language and clarity.

23. Line 94 and elsewhere throughout the text - "rescuing or confiscating news" is a bit confusing term (it suggests that news are rescuing and confiscating something), I would suggest replacing it throughout the text with "rescue or confiscation related news"

24. All species should be presented with both their scientific and vernacular name when they are mentioned in the text for the first time, and afterwards either scientific or vernacular names should be consistently used throughout the text. For example, in Methods (Lines 99-103) only Latin names are presented, while in the Discussion all species are referred to only by their vernacular name.

25. Line 127 - there is one equals sign out of place there (the second one)

26. Line 129 - variable pi for Shannon-Wiener Index is incorrectly defined here, it does not represent the proportion of S comprising the ith species, but the proportion of individuals belonging to ith species.

27. Line 135 - replace "We could filter" with "We filtered"

28. Lines 140-146 - this section is very confusing. I don't understand the meaning of rescue events being reported "in 10 years", "in seven years". Did the authors want to say that they were recorded only in that many of the total number of studied years? If that is so, it is not really informative and could be perhaps omitted - it would be more informative to discuss here about the number of news articles per species, instead of years.

29. Lines 145-146 - replace "to be mounting" with "to increase", and rephrase "appeared to be more fluctuated"

30. Line 151 - replace "during seventeen years" with "over the studied period". Also, use numbers if a number is larger than 10.

31. Line 155, "significantly high" - a value can not be significant by itself, only in comparison with another value, so either state just "high" or "significantly higher than" and compare it with another value.

32. Line 177, "Public recognition of species" - is it really a general"public", if the one making identification was an official person, or a journalist? I would suggest shortening the title to just "Species recognition".

33. Line 178 - replace "could be firstly recognized as" with "were reported as"

34. Line 181 - "there were no variations in recognitions on..." should be replaced with "the recognition of .... was consistent"

35. Line 185, "Only recognition of "species"" - I do not understand this, nor the use of quotation marks.

36. Line 191 - replace "occurred in species" with "were related to species"

37. Line 236 - replace "of smuggling" with "for smuggling"

38. Line 241 - delete "might"

39. Line 244 - delete "away"

40. Line 257 - delete the second comma, and replace "and were" with "and the two species were"

41. Line 266 - I would suggest inserting "might experience elevated mortality and" before the word "perish"

42. Line 269 - delete "oftentimes", and replace "was" with "is often"

43. Line 275-276 - I would suggest mentioning here also the problem of introduced specimens threatening local populations as disease vectors, and referring to some relevant references.

44. Line 281 - replace "Along" with "As"

45. Lines 287-288 - replace "emphasised" with "emphasize", "on some certain" with "of some", and "called" with "calls"

46. Line 300 - replace "their" with "them"; I am not sure if "Chinese people" is a proper term

47. Line 304 - delete "its"

48. Lines 305-306 - this sentence is unclear, there seems to be more Chinese terms than translations.

49. Line 322 - delete "Fundamentally,"

50. Lines 329-330 - replace "high ranked protecting category" with "high threat level" or "high threat category", and "less public knowledge" with either "low public awareness" or "poor public knowledge"

51. Line 330 - delete "extremely"

52. Line 334 - delete "certain"

53. Whole Conclusion section needs thorough language editing

54. Line 336 - replace " 'invisible' " with "unrecognized"

55. Line 337 - replace "involved" with "used"

56. Line 338, "making the topic regarding captive management less sufficient" - please rephrase

57. Line 338 - Start the sentence with "Nevertheless, we hope..."

58. Line 341 - replace "managements" with "management"


Figures and tables:

59. Figure 1 - units are missing in the figure legend (i.e. "number of species", "number of reports", etc.)

60. Figure 2 caption - replace "total" with the "the total number of", "in each" with "per each", "Databse" with "Database"

61. Figure 3 caption - replace "Annual change of" with "Annual trends in"

62. Figure 4 caption - delete ""Significant"

63. Table 1 caption - insert "studied" before the word "periods"

64. Table 2 - insert commas and spaces between terms in the third column


References:

65. Reference Reuter & Schaefer (2016), cited in Lines 44 and 52, is missing from the reference list.

66. Order of citations in the text where multiple references are listed should be checked and corrected throughout the text, they should be ordered first chronologically and then alphabetically (e.g. see Line 70)

67. Line 254 - one of the authors is cited here as Wuester, while in the reference list it is cited as Wüster

68. Line 281 - one of the authors is cited here as Wisweda, while in the reference list it is cited as Wiswedel

69. In Line 328, there is a reference Nekaris et al. (2014) which is missing from the reference list, while in the reference list there is a reference Nekaris (2014) (Line 424) which is not cited in the text. Should that be the same sentence

70. Reference Nekaris and Starr 2015 is listed twice in reference list, in Lines 426 and 431

71. References in the reference list should be reordered, so they are listed first alphabetically, then for the same first author to list first single-author publications, followed by two-authored and finally multiple-author papers, and then chronologically.

72. Some journals names are abbreviated (e.g. Line 354), some lack proper capitalization (e.g. Line 351)

73. Line 469 - "cites" should be capitalized

Reviewer 2 ·

Basic reporting

The MS is written well, although there are some minor grammar issues throughout that should be very easy to fix that would improve the clarity (e.g. line 41 'many these species' should be 'many of').

The referencing is done well but in places there are statements made where citations should be be added, or where referencing could be improved:
L47/48: provide reference that shows that IWT can remove countries' control of natural resources
L58, 66 and 120: correctly reference the UNEP WCMC Trade Database where figures from the trade data are used (in L58 this might be a reference to the following sentence's citation, if so, include it earlier too) - citation for Trade Database should be as suggested here: https://trade.cites.org/cites_trade_guidelines/en-CITES_Trade_Database_Guide.pdf
L80-81: I am not sure if media attention always promotes conservation, and this paper doesn't seem to say that directly. I would suggest finding another reference here.

Figures
Figure 2 is confusing - does it mean the % of total CITES trade reported? I also think the two panels are hard to compare because the species are in order of proportion, so the order is different for each panel. A different format might be more appropriate - maybe a stacked bar with both panels combined into it?

Figure 3: suggest changing the colours in C so they are different to B, to clearly show that it is different taxa and not still individuals and events, as reading quickly down the graphs makes it unclear.

Figure 4: I found this very confusing to read, and I wasn't sure why these two things were reported together, and I am not clear what the take away message is from this figure at all.

Experimental design

L96: Please give more details on your search protocol. For example, when did you carry out your search? Did you use all results or did you limit it in anyway? L104 What was your definition of a news article? Was it only formal news outlets, or informal blogs etc too? Also 104: how did you define rescue events? Did you use extra keywords for this, or did you sort the results by hand to judge which were rescue events?

L119: My main concern with the MS is that the methods around the CITES Trade Database analysis have not been described in any detail. In order for a reader to know what you have done, and for anybody to replicate it, the following information needs to be included when you describe your analysis: when did you download data?; which Terms, Units and Sources did you use in your analysis?; Did you look at importer or exporter reported data, or a combination of both? And if the latter, if exporter and importer reported data differed, which did you go with?; Did you omit re-exports from your analysis?; Did you look only at imports into China (I ask, because there are raw data labelled exports too - does this mean exporter reported data of exports to China from other countries?).

Validity of the findings

There are some conclusions that are not supported by the data collected, and more that need some strengthening.

L73/74 and L283: CITES does not cover national or illegal trade, and in fact this is clearly not its role. In particular, references to how CITES is not doing anything about national trade should be reworded, because that's right, it shouldn't be. It should pick up regional trade (apart from in the EU), although it will still only be records of legal trade. Your whole discussion section on how better data are needed to inform conservation is a good point, but it is undermined by the use of CITES as an example, as you have not focussed solely on seizures of primates as they enter the country (which would show a discrepancy with the CITES data). You also don't link your methods of monitoring seizures via news reports to this - are you suggesting that this is a way of collecting these data instead?

L200: This conclusion should be toned down - if it is not reported on internet news this does not mean that it has not been recorded in official data by the Chinese authorities - there are likely to be many seizures not reported in the news for several reasons. If you want to keep this point in then I would suggest a brief discussion of potential biases of news reports here unless there are official statistics available that can be cited.

L307-309: You measured seizures and confiscations, which surely is a sign of effective enforcement? Unless there is evidence that there are many more primates in trade that are not being seized, I don't think you can draw this conclusion from your findings. While I am sure massive IWT in primates is the case, you should strengthen this argument.

L315-317: Awareness raising is not the same as demand reduction, and simply raising awareness is not gong to automatically lead to behaviour change in consumers (e.g. see diagram here: Breaking the Brand (2016). https://breakingthebrand.org/how-much-is-spent-on-rhino-horn-demand-reduction-campaigns/?doing_wp_cron=1534087399.6144099235534667968750

L338: I think there are more limitations to your study apart from this one, and I think these should be discussed in more detail earlier in the MS. I have highlighted most of these already earlier in the review (biases of news reports, for example).

Reviewer 3 ·

Basic reporting

This is an interesting study about the temporal and spatial trends of primate trade in China over the last 18 years using news reports from the internet. The authors estimated national primate trade and public awareness towards primate rescue and confiscation by quantifying the number of native species uncovered in unique web news from 2000-2017. This is potentially an important study in the area of illegal wildlife trade and one that provides some evidence that an alternative estimate to the illegal trade might be obtained using the web. The writing is clear and there is sufficient context provided. Nevertheless, the authors will need to provide some clarifications to their existing methodology and I also provided some suggestions to improve the manuscript.

Experimental design

Some issues with the methods are listed below.

The source of the reports is not available anywhere. I thought that is critical and must be included in the supplementary materials. For instance, provide a link to each news report.

Line 82. The authors should describe how public knowledge is quantified by looking at the comments and data. The exact approach is unclear. This will affect the other data such as public recognition of the species.

Line 107. How did the authors evaluate the physical condition of rescued individuals? If the information is gleaned from the news, which is the authority (e.g., vet) that did it?

Line 126. Again, be clear on how public knowledge is measured.

Line 127. In this sentence, it is not clear what the diversity indices are used for.

Validity of the findings

The validity of the findings provided will depend on the methods used, which at this moment is quite unclear. But at this moment, the findings seem fine.

The data fills an important gap in the illegal wildlife trade in China. More data is coming out from the internet or online and this research is I believe one of the first ones to present information on the Chinese primate trade in the last decade .

---

## Round 0.2 · Minor Revisions

Many thanks for your careful consideration of the previous reviewers' comments. There are only a few minor points raised but the current reviewer and this I believe can easily be made. I look forward to recommending acceptance of the manuscript.

Reviewer 2 ·

Basic reporting

No comment

Experimental design

The text has been strengthened well and the clarification of the methodology has been really helpful. The reporting of the CITES Trade Database analysis that I particularly highlighted has also been improved, with only the following points of clarification:

L63 - you say 11,000 other individuals traded as body parts - unless you have applied some kind of conversion factor here to translate parts into whole organisms (as is done in Harfoot et al. 2018 Unveiling the patterns and trends in 40 years of global trade in CITES-listed wildlife. Biological Conservation) then you can't make the link between individual body parts and individual animals. I you have converted it then mention this. It is a minor point, but could confuse readers.

In addition, Table 1 suggests that some of the species are not listed in the CITES Appendices, but there is an Order listing for primates, and several Family listings so I am not sure if this is correct - species listings can be checked here (unless they are unlisted synonyms or new species): https://www.speciesplus.net/species

Validity of the findings

No comment

Additional comments

The text has been greatly improved since the first submission, and it is now much clearer. I am also impressed by the approach that the authors have taken to incorporating references where suggested, and I think this has strengthened their points. The figures look great, and I think that the new figure 1 is really useful for understanding the methodology. Finally, and most importantly, the conclusions are now much better linked to the findings.

---

## Round 0.3 · accepted · Accept

Many thank you for considering the comments in the latest round of reviews and the previous reviews. I'm happy to recommend that the manuscript now be accepted

#